# Spectral Mixture Kernels for Multi-Output Gaussian Processes

**Gabriel Parra**
Department of Mathematical Engineering
Universidad de Chile
gparra@dim.uchile.cl

**Felipe Tobar**
Center for Mathematical Modeling
Universidad de Chile
ftobar@dim.uchile.cl

## Abstract

Early approaches to multiple-output Gaussian processes (MOGPs) relied on linear combinations of independent, latent, single-output Gaussian processes (GPs). This resulted in cross-covariance functions with limited parametric interpretation, thus conflicting with the ability of single-output GPs to understand lengthscales, frequencies and magnitudes to name a few. On the contrary, current approaches to MOGP are able to better interpret the relationship between different channels by directly modelling the cross-covariances as a spectral mixture kernel with a phase shift. We extend this rationale and propose a parametric family of complex-valued cross-spectral densities and then build on Cramér's Theorem (the multivariate version of Bochner's Theorem) to provide a principled approach to design multivariate covariance functions. The so-constructed kernels are able to model delays among channels in addition to phase differences and are thus more expressive than previous methods, while also providing full parametric interpretation of the relationship across channels. The proposed method is first validated on synthetic data and then compared to existing MOGP methods on two real-world examples.

## 1 Introduction

The extension of Gaussian processes (GPs [1]) to multiple outputs is referred to as multi-output Gaussian processes (MOGPs). MOGPs model temporal or spatial relationships among infinitely-many random variables, as scalar GPs, but also account for the statistical dependence *across* different sources of data (or channels). This is crucial in a number of real-world applications such as fault detection, data imputation and denoising. For any two input points $x, x'$, the covariance function of an $m$-channel MOGP $k(x, x')$ is a symmetric positive-definite $m \times m$ matrix of scalar covariance functions. The design of this matrix-valued kernel is challenging since we have to deal with the trade off between (i) choosing a broad class of $m(m-1)/2$ cross-covariances and $m$ auto-covariances, while at the same time (ii) ensuring positive definiteness of the symmetric matrix containing these $m(m+1)/2$ covariance functions for any pair of inputs $x, x'$. In particular, unlike the widely available families of auto-covariance functions (e.g., [2]), cross-covariances are not bound to be positive definite and therefore can be designed freely; the construction of these functions with interpretable functional form is the main focus of this article.

A classical approach to define cross-covariances for a MOGP is to linearly combine independent latents GPs, this is the case of the Linear Model of Coregionalization (LMC [3]) and the Convolution Model (CONV, [4]). In these cases, the resulting kernel is a function of both the covariance functions of the latent GPs and the parameters of the linear operator considered; this results in symmetric and centred cross-covariances. While these approaches are simple, they lack interpretability of the dependencies learnt and force the auto-covariances to have similar behaviour across different channels. The LMC method has also inspired the Cross-Spectral Mixture (CSM) kernel [5], which uses the

Spectral Mixture (SM) kernel in [6] within LMC and model phase differences across channels by manually introducing a shift between the cosine and exponential factors of the SM kernel. Despite exhibiting improved performance wrt previous approaches, the addition of the shift parameter in CSM poses the following question: Can the spectral design of multiouput covariance functions be even more flexible?

We take a different approach to extend the spectral mixture concept to multiple outputs: Recall that for stationary scalar-valued GPs, [6] designs the power spectral density (PSD) of the process by a mixture of square exponential functions to then, supported by Bochner's theorem [7], present the Spectral Mixture kernel via the inverse Fourier transform of the so-constructed PSD. Along the same lines, our main contribution is to propose an expressive family of complex-valued square-exponential cross-spectral densities, and then build on Cramér's theorem [8, 9], the multivariate extension of Bochner's, to construct the Multi-Output Spectral Mixture kernel (MOSM). The proposed multivariate covariance function accounts for all the properties of the Cross-Spectral Mixture kernel in [5] plus a delay component across channels and variable parameters for auto-covariances of different channels. Additionally, the proposed MOSM provides clear interpretation of all the parameters in spectral terms. Our experimental contribution includes an illustrative example using a trivariate synthetic signal and validation against all the aforementioned literature using two real-world datasets.

## 2 Background

**Definition 1.** *A Gaussian process (GP) over the input set $\mathcal{X}$ is a real-valued stochastic process $(f(x))_{x \in \mathcal{X}}$ such that for any finite subset of inputs $\{x_i\}_{i=1}^N \subset \mathcal{X}$, the random variables $\{f(x_i)\}_{i=1}^N$ are jointly Gaussian. Without loss of generality we will choose $\mathcal{X} = \mathbb{R}^n$.*

A GP [1] defines a distribution over functions $f(x)$ that is uniquely determined by its mean function $m(x) := \mathbb{E}(f(x))$, typically assumed $m(x) = 0$, and its covariance function (also known as kernel) $k(x, x') := \text{cov}(f(x), f(x'))$, $x, x' \in \mathcal{X}$. We now equip the reader with the necessary background to follow our proposal: we first review a spectral-based approach to the design of scalar-valued covariance kernels and then present the definition of a multi-output GP.

### 2.1 The Spectral Mixture kernel

To bypass the explicit construction of positive-definite functions within the design of stationary covariance kernels, it is possible to design the power spectral density (PSD) instead [6] and then transform it into a covariance function using the inverse Fourier transform. This is motivated by the fact that the strict positivity requirement of the PSD is much easier to achieve than the positive definiteness requirement of the covariance kernel. The theoretical support of this construction is given by the following theorem:

**Theorem 1.** *(Bochner's theorem) An integrable[1] function $k : \mathbb{R}^n \to \mathbb{C}$ is the covariance function of a weakly-stationary mean-square-continuous stochastic process $f : \mathbb{R}^n \to \mathbb{C}$ if and only if it admits the following representation*

$$k(\tau) = \int_{\mathbb{R}^n} e^{\iota \omega^\top \tau} S(\omega) \mathrm{d}\omega \tag{1}$$

*where $S(\omega)$ is a non-negative bounded function on $\mathbb{R}^n$ and $\iota$ denotes the imaginary unit.*

For a proof see [9]. The above theorem gives an explicit relationship between the spectral density $S$ and the covariance function $k$ of the stochastic process $f$. In this sense, [6] proposed to model the spectral density $S$ as a weighted mixture of $Q$ square-exponential functions, with weights $w_q$, centres $\mu_q$ and diagonal covariance matrices $\Sigma_q$, that is,

$$S(\omega) = \sum_{q=1}^Q w_q \frac{1}{(2\pi)^{n/2}|\Sigma_q|^{1/2}} \exp\left(-\tfrac{1}{2}(\omega - \mu_q)^\top \Sigma_q^{-1}(\omega - \mu_q)\right). \tag{2}$$

Relying on Theorem 1, the kernel associated to the spectral density $S(\omega)$ in eq. (2) is given the spectral mixture kernel defined as follows.

**Definition 2.** *A Spectral Mixture (SM) kernel is a positive-definite stationary kernel given by*

$$k(\tau) = \sum_{q=1}^{Q} w_q \exp\left(-\frac{1}{2}\tau^\top \Sigma_q \tau\right) \cos(\mu_q^\top \tau) \tag{3}$$

*where $\mu_q \in \mathbb{R}^n$, $\Sigma_q = \mathrm{diag}(\sigma_1^{(q)}, \ldots, \sigma_n^{(q)})$ and $w_q, \sigma_q \in \mathbb{R}_+$.*

Due to the universal function approximation property of the mixtures of Gaussians (considered here in the frequency domain) and the relationship given by Theorem 1, the SM kernel is able to approximate continuous stationary kernels to an arbitrary precision given enough spectral components as is [10, 11]. This concept points in the direction of sidestepping the kernel selection problem in GPs and it will be extended to cater for multivariate GPs in Section 3.

### 2.2 Multi-Output Gaussian Processes

A multivariate extension of GPs can be constructed by considering an ensemble of scalar-valued stochastic processes where any finite collection of values across all such processes are jointly Gaussian. We formalise this definition as follows.

**Definition 3.** *An $m$-channel multi-output Gaussian process $\mathbf{f}(x) := (f_1(x), \ldots, f_m(x))$, $x \in \mathcal{X}$, is an $m$-tuple of stochastic processes $f_p : \mathcal{X} \to \mathbb{R} \,\forall p = 1, \ldots, m$, such that for any (finite) subset of inputs $\{x_i\}_{i=1}^N \subset \mathcal{X}$, the random variables $\{f_{c(i)}(x_i)\}_{i=1}^N$ are jointly Gaussian for any choice of indices $c(i) \in \{1, \ldots, m\}$.*

Recall that the construction of scalar-valued GPs requires choosing a scalar-valued mean function and a scalar-valued covariance function. Conversely, an $m$-channel MOGP is defined by an $\mathbb{R}^m$-valued mean function, whose $i^{\text{th}}$ element denotes the mean function of the $i^{\text{th}}$ channel, and an $\mathbb{R}^m \times \mathbb{R}^m$-valued covariance function, whose $(i, j)^{\text{th}}$ element denotes the covariance between the $i^{\text{th}}$ and $j^{\text{th}}$ channels. The symmetry and positive-definiteness conditions of the MOGP kernel are defined as follows.

**Definition 4.** *A two-input matrix-valued function $\mathcal{K}(x, x') : \mathcal{X} \times \mathcal{X} \to \mathbb{R}^{m \times m}$ defined element-wise by $[\mathcal{K}(x, x')]_{ij} = k_{ij}(x, x')$ is a multivariate kernel (covariance function) if it is:*

*(i) Symmetric, i.e., $\mathcal{K}(x, x') = \mathcal{K}(x', x)^\top, \forall x, x' \in \mathcal{X}$, and*

*(ii) Positive definite, i.e., $\forall N \in \mathbb{N}, \mathbf{c} \in \mathbb{R}^{N \times m}, \mathbf{x} \in \mathcal{X}^N$ such that, $[\mathbf{c}]_{pi} = c_{pi}, [\mathbf{x}]_p = x_p$, we have*

$$\sum_{i,j=1}^{m} \sum_{p,q=1}^{N} c_{pi} c_{qj} k_{ij}(x_p, x_q) \geq 0. \tag{4}$$

Furthermore, we say that a multivariate kernel $\mathcal{K}(x, x')$ is stationary if $\mathcal{K}(x, x') = \mathcal{K}(x - x')$ or equivalently $k_{ij}(x, x') = k_{ij}(x - x') \,\forall i, j \in \{1, \ldots, m\}$, in this case, we denote $\tau = x - x'$.

The design of the MOGP covariance kernel involves jointly choosing functions that model the covariance of each channel (diagonal elements in $\mathcal{K}$) and functions that model the cross-covariance between different channels at different input locations (off-diagonal elements in $\mathcal{K}$). Choosing these $m(m + 1)/2$ covariance functions is challenging when we want to be as expressive as possible and include, for instance, delays, phase shifts, negative correlations or to enforce specific spectral content while at the same time maintaining positive definiteness of $\mathcal{K}$. The reader is referred to [12, 13] for a comprehensive review of MOGP models.

## 3 Designing Multi-Output Gaussian Processes in the Fourier Domain

We extend the spectral-mixture approach [6] to multi-output Gaussian processes relying on the multivariate version of Theorem 1 first proved by Cramér and thus referred to as Cramér's Theorem [8, 9] given by

**Theorem 2.** *(Cramér's Theorem) A family $\{k_{ij}(\tau)\}_{i,j=1}^m$ of integrable functions are the covariance functions of a weakly-stationary multivariate stochastic process if and only if they (i) admit the*

*representation*

$$k_{ij}(\tau) = \int_{\mathbb{R}^n} e^{\iota \omega^\top \tau} S_{ij}(\omega) \mathrm{d}\omega \qquad \forall i, j \in \{1, \ldots, m\} \tag{5}$$

*where each $S_{ij}$ is an integrable complex-valued function $S_{ij} : \mathbb{R}^n \to \mathbb{C}$ known as the spectral density associated to the covariance function $k_{ij}(\tau)$, and (ii) fulfil the positive definiteness condition*

$$\sum_{i,j=1}^{m} \overline{z_i} z_j S_{ij}(\omega) \geq 0 \qquad \forall \{z_1, \ldots, z_m\} \subset \mathbb{C}, \ \omega \in \mathbb{R}^n \tag{6}$$

*where $\overline{z}$ denotes the complex conjugate of $z \in \mathbb{C}$.*

Note that eq. (5) states that each covariance function $k_{ij}$ is the inverse Fourier transform of a spectral density $S_{ij}$, therefore, we will say that these functions are *Fourier pairs*. Accordingly, we refer to the set of arguments of the covariance function $\tau \in \mathbb{R}^n$ as *time* or *space Domain* depending of the application considered, and to the set of arguments of the spectral densities $\omega \in \mathbb{R}^n$ as *Fourier* or *spectral domain*. Furthermore, a direct consequence of the above theorem is that for any element $\omega$ in the Fourier domain, the matrix defined by $S(\omega) = [S_{ij}(\omega)]_{i,j=1}^m \in \mathbb{R}^{m \times m}$ is Hermitian, i.e., $S_{ij}(\omega) = \overline{S}_{ji}(\omega) \ \forall i, j, \omega$.

Theorem 2 gives the guidelines to construct covariance functions for MOGP by designing their corresponding spectral densities instead, i.e., the design is performed in the Fourier rather than the space domain. The simplicity of design in the Fourier domain stems from the positive-definiteness condition of the spectral densities in eq. (6), which is much easier to achieve than that of the covariance functions in eq. (4). This can be understood through an analogy with the univariate model: in the single-output case the positive-definiteness condition of the kernel only requires positivity of the spectral density, whereas in the multioutput case the positive-definiteness condition of the multivariate kernel only requires that the matrix $S(\omega)$, $\forall \omega \in \mathbb{R}^n$, is positive definite but there are no constraints on each function $S_{ij} : \omega \mapsto S_{ij}(\omega)$.

## 3.1 The Multi-Output Spectral Mixture kernel

We now propose a family of Hermitian positive-definite complex-valued functions $\{S_{ij}(\cdot)\}_{i,j=1}^m$, thus fulfilling the requirements of Theorem 2, eq. (6), to use them as cross-spectral densities within MOGP. This family of functions is designed with the aim of providing physical parametric interpretation and closed-form covariance functions after applying the inverse Fourier transform.

Recall that complex-valued positive-definite matrices can be decomposed in the form $S(\omega) = R^H(\omega) R(\omega)$, meaning that the $(i, j)^{\text{th}}$ entry of $S(\omega)$ can be expressed as $S_{ij}(\omega) = R_{:i}^H(\omega) R_{:j}(\omega)$; where $R(\omega) \in \mathbb{C}^{Q \times m}$, $R_{:i}(\omega)$ is the $i^{\text{th}}$ column of $R(\omega)$, and $(\cdot)^H$ denotes the Hermitian (transpose and conjugate) operator. Note that this factor decomposition fulfils eq. (6) for any choice of $R(\omega) \in \mathbb{C}^{Q \times m}$:

$$\sum_{i,j=1}^{m} \overline{z_i} R_{:i}^H(\omega) R_{:j}(\omega) z_j = \left|\left| \sum_{i=1}^{m} z_i R_{:i}(\omega) \right|\right|^2 = ||R(\omega)\mathbf{z}||^2 \geq 0 \ \forall \mathbf{z} = [z_1, \ldots, z_m]^\top \in \mathbb{C}^m, \ \omega \in \mathbb{R}^n \tag{7}$$

We refer to $Q$ as the rank of the decomposition, since by choosing $Q < m$ the rank of $S(\omega) = R^H(\omega) R(\omega)$ can be at most $Q$. For ease of notation we choose[2] $Q = 1$, where the columns of $R(\omega)$ are complex-valued functions $\{R_i\}_{i=1}^m$, and $S(\omega)$ is modeled as a rank-one matrix according to $S_{ij}(\omega) = \overline{R}_i(\omega) R_j(\omega)$. Since Fourier transforms and multiplications of square exponential (SE) functions are also SE, we model $R_i(\omega)$ as a complex-valued SE function so as to ensure closed-form expression of its corresponding covariance kernel, that is,

$$R_i(\omega) = w_i \exp\left( -\frac{1}{4}(\omega - \mu_i)^\top \Sigma_i^{-1}(\omega - \mu_i) \right) \exp\left( -\iota(\theta_i^\top \omega + \phi_i) \right), \quad i = 1, \ldots, m \tag{8}$$

where $w_i, \phi_i \in \mathbb{R}$, $\mu_i, \theta_i \in \mathbb{R}^n$ and $\Sigma_i = \mathrm{diag}([\sigma_{i1}^2, \ldots, \sigma_{in}^2]) \in \mathbb{R}^{n \times n}$. With this choice of the functions $\{R_i\}_{i=1}^m$, the spectral densities $\{S_{ij}\}_{i,j=1}^m$ are given by

$$S_{ij}(\omega) = w_{ij} \exp\left( -\frac{1}{2}(\omega - \mu_{ij})^\top \Sigma_{ij}^{-1}(\omega - \mu_{ij}) + \iota(\theta_{ij}^\top \omega + \phi_{ij}) \right), \quad i, j = 1, \ldots, m \tag{9}$$

meaning that the cross-spectral density between channels $i$ and $j$ is modeled as a complex-valued SE function with the following parameters:

- covariance: $\Sigma_{ij} = 2\Sigma_i(\Sigma_i + \Sigma_j)^{-1}\Sigma_j$
- mean: $\mu_{ij} = (\Sigma_i + \Sigma_j)^{-1}(\Sigma_i\mu_j + \Sigma_j\mu_i)$
- magnitude: $w_{ij} = w_i w_j \exp\left(-\frac{1}{4}(\mu_i - \mu_j)^\top(\Sigma_i + \Sigma_j)^{-1}(\mu_i - \mu_j)\right)$
- delay: $\theta_{ij} = \theta_i - \theta_j$
- phase: $\phi_{ij} = \phi_i - \phi_j$

where the so-constructed magnitudes $w_{ij}$ ensure positive definiteness and, in particular, the auto-spectral densities $S_{ii}$ are real-valued SE functions (since $\theta_{ii} = \phi_{ii} = 0$) as in the standard (scalar-valued) spectral mixture approach [6].

The power spectral density in eq. (9) corresponds to a complex-valued kernel and therefore to a complex-valued GP [14, 15] . In order to restrict this generative model only to real-valued GPs, the proposed power spectral density has to be symmetric with respect to $\omega$ [16], we then make $S_{ij}(\omega)$ symmetric simply by reassigning $S_{ij}(\omega) \mapsto \frac{1}{2}(S_{ij}(\omega) + S_{ij}(-\omega))$, this is equivalent to choosing $R_i(\omega)$ to be a vector of two *mirrored* complex SE functions.

The resulting (symmetric with respect to $\omega$) cross-spectral density between the $i^{\text{th}}$ and $j^{\text{th}}$ channels $S_{ij}(\omega)$ and its corresponding real-valued kernel $k_{ij}(\tau) = \mathcal{F}^{-1}\{S_{ij}(\omega)\}(\tau)$ are the following Fourier pairs

$$S_{ij}(\omega) = \frac{w_{ij}}{2}\left(e^{\left(\frac{-1}{2}(\omega-\mu_{ij})^\top\Sigma_{ij}^{-1}(\omega-\mu_{ij})+\iota\left(\theta_{ij}^\top\omega+\phi_{ij}\right)\right)} + e^{\left(\frac{-1}{2}(\omega+\mu_{ij})^\top\Sigma_{ij}^{-1}(\omega+\mu_{ij})+\iota\left(-\theta_{ij}^\top\omega+\phi_{ij}\right)\right)}\right)$$

$$k_{ij}(\tau) = \alpha_{ij}\exp\left(-\frac{1}{2}(\tau+\theta_{ij})^\top\Sigma_{ij}(\tau+\theta_{ij})\right)\cos\left((\tau+\theta_{ij})^\top\mu_{ij}+\phi_{ij}\right) \tag{10}$$

where the magnitude parameter $\alpha_{ij} = w_{ij}(2\pi)^{\frac{n}{2}}|\Sigma_{ij}|^{1/2}$ absorbs the constant resulting from the inverse Fourier transform.

We can again confirm that the autocovariances ($i = j$) are real-valued and contain square-exponential and cosine factors as in the scalar SM approach since $\alpha_{ii} \geq 0$ and $\theta_{ii} = \phi_{ii} = 0$. Conversely, the proposed model for the cross-covariance between different channels ($i \neq j$) allows for (i) both negatively- and positively-correlated signals ($\alpha_{ij} \in \mathbb{R}$), (ii) delayed channels through the delay parameter $\theta_{ij} \neq 0$ and (iii) out-of-phase channels where the covariance is not symmetric with respect to the delay for $\phi_{ij} \neq 0$. Fig. 1 shows cross-spectral densities and their corresponding kernel for a choice of different delay and phase parameters.

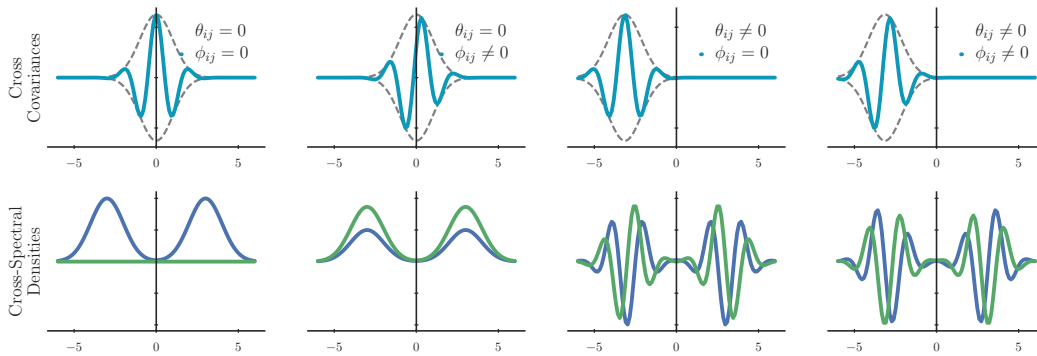

Figure 1: Power spectral density and kernels generated by the proposed model in eq. (10) for different parameters. **Bottom**: Cross-spectral densities, real part in blue and imaginary part in green. **Top**: Cross-covariance functions in blue with reference SE envelope in dashed line. **From left to right**: zero delay and zero phase; zero delay and non-zero phase; non-zero delay and zero phase; and non-zero delay and non-zero phase.

The kernel in eq. (10) resulted from a low rank choice for the PSD matrix $S_{ij}$, therefore, increasing the rank in the proposed model for $S_{ij}$ is equivalent to consider several kernel components. Arbitrarily choosing $Q$ of these components yields the expression for the proposed multivariate kernel:

**Definition 5.** *The Multi-Output Spectral Mixture kernel (MOSM) has the form:*

$$k_{ij}(\tau) = \sum_{q=1}^{Q} \alpha_{ij}^{(q)} \exp\left(-\frac{1}{2}(\tau + \theta_{ij}^{(q)})^{\top}\Sigma_{ij}^{(q)}(\tau + \theta_{ij}^{(q)})\right) \cos\left((\tau + \theta_{ij}^{(q)})^{\top}\mu_{ij}^{(q)} + \phi_{ij}^{(q)}\right) \quad (11)$$

*where $\alpha_{ij}^{(q)} = w_{ij}^{(q)}(2\pi)^{\frac{n}{2}}|\Sigma_{ij}^{(q)}|^{1/2}$ and the superindex $(\cdot)^{(q)}$ denotes the parameter of the $q^{th}$ component of the spectral mixture.*

This multivariate covariance function has spectral-mixture positive-definite kernels as auto-covariances, while the cross-covariances are spectral mixture functions with different parameters for different output pairs, which can be (i) non-positive-definite, (ii) non-symmetric, and (iii) delayed with respect to one another. Therefore, the MOSM kernel is a multi-output generalisation of the spectral mixture approach [6] where the positive definiteness is guaranteed by the factor decomposition of $S_{ij}$ as shown in eq. (7).

## 3.2 Training the model and computing the predictive posterior

Fitting the model to observed data follows the same rationale of standard GP, that is, maximising log-probability of the data. Recall that the observations in the multioutput case consist of (i) a location $x \in \mathcal{X}$, (ii) a channel identifier $i \in \{1, \dots, m\}$, and (iii) an observed value $y \in \mathbb{R}$; therefore, we denote $N$ observations as the set of 3-tuples $D = \{(x_c, i_c, y_c)\}_{c=1}^{N}$. As all observations are jointly Gaussian, we concatenate the observations into the three vectors $\mathbf{x} = [x_1, \dots, x_N]^{\top} \in \mathcal{X}^N$, $\mathbf{i} = [i_1, \dots, i_N]^{\top} \in \{1, \dots, m\}^N$, and $\mathbf{y} = [y_1, \dots, y_N]^{\top} \in \mathbb{R}^N$, to express the negative log-likelihood (NLL) by

$$-\log p(\mathbf{y}|\mathbf{x}, \Theta) = \frac{N}{2}\log 2\pi + \frac{1}{2}\log|\mathbf{K}_{\mathbf{xi}}| + \frac{1}{2}\mathbf{y}^{\top}\mathbf{K}_{\mathbf{xi}}^{-1}\mathbf{y} \quad (12)$$

where all hyperparameters are denoted by $\Theta$, and $\mathbf{K}_{\mathbf{xi}}$ is the covariance matrix of all observed samples, that is, the $(r, s)^{\text{th}}$ element $[\mathbf{K}_{\mathbf{xi}}]_{rs}$ is the covariance between the process at (location: $x_r$, channel: $i_r$) and the process at (location: $x_s$, channel: $i_s$). Recall that, under the proposed MOSM model, this covariance $[\mathbf{K}_{\mathbf{xi}}]_{rs}$ is given by eq. (11), that is, $k_{i_r i_s}(x_r - x_s) + \sigma_{i_r,\text{noise}}^2 \delta_{i_r i_s}$, where $\sigma_{i_r,\text{noise}}^2$ is a diagonal term to cater for uncorrelated observation noise. The NLL is then minimised with respect to $\Theta = \{w_i^{(q)}, \mu_i^{(q)}, \Sigma_i^{(q)}, \theta_i^{(q)}, \phi_i^{(q)}, \sigma_{i,\text{noise}}^2\}_{i=1,q=1}^{m,Q}$, that is, the original parameters chosen to construct $R(\omega)$ in Section 3.1, plus the noise hyperparameters.

Once the hyperparameters are optimised, computing the predictive posterior in the proposed MOSM follows the standard GP procedure with the joint covariances given by eq. (11).

## 3.3 Related work

Generalising the scalar spectral mixture kernel to MOGPs can be achieved from the LMC framework as pointed out in [5] (denoted SM-LMC). As this formulation only considers real-valued cross spectral densities, the authors propose a multivariate covariance function by including a complex component to the cross spectral densities to cater for phase differences across channels, which they call the Cross Spectral Mixture kernel (denoted CSM). This multivariate covariance function can be seen as the proposed MOSM model with $\mu_i = \mu_j$, $\Sigma_i = \Sigma_j$, $\theta_i = \theta_j$ $\forall i, j \in \{1, \dots, m\}$ and $\phi_i = \mu_i^{\top}\psi_i$ for $\psi_i \in \mathbb{R}^n$. As a consequence, the SM-LMC is a particular case of the proposed MOSM model, where the parameters $\mu_i, \Sigma_i, \theta_i$ are restricted to be same for all channels and therefore no phase shifts and no delays are allowed—unlike the MOSM example in Fig. 1. Additionally, Cramér's theorem has also been used in a similar fashion in [17] but only with real-valued t-Student cross-spectral densities yielding cross-covariances that are either positive-definite or negative-definite.

# 4 Experiments

We show two sets of experiments. First, we validated the ability of the proposed MOSM model in the identification of known auto- and cross-covariances of synthetic data. Second, we compared

MOSM against the spectral-mixture linear model of coregionalization (SM-LMC, [3, 6, 5]), the Gaussian convolution model (CONV, [4]), and the cross-spectral mixture model (CSM, [5]) in the estimation of missing real-world data in two different distributed settings: climate signals and metal concentrations. All models were implemented in Tensorflow [18] using GPflow [19] in order to make use of automatic differentiation to compute the gradients of the NLL. The performance of all the models in the experiments was measured by the mean absolute error given by

$$\text{MAE} : \frac{1}{N} \sum_{i=1}^{N} |y_i - \hat{y}_i| \tag{13}$$

where $y_i$ denotes the true value and $\hat{y}_i$ the MOGP estimate.

## 4.1  Synthetic example: Learning derivatives and delayed signals

All models were implemented to recover the auto- and cross-covariances of a three-output GP with the following components: (i) a reference signal sampled from a GP $f(x) \sim \mathcal{GP}(0, K_{SM})$ with spectral mixture covariance kernel $K_{SM}$ and zero mean, (ii) its derivative $f'(x)$, and (iii) a delayed version $f_\delta(x) = f(x - \delta)$. The motivation for this illustrative example is that the covariances and cross covariances of the aforementioned processes are known explicitly (see [1, Sec. 9.4]) and we can therefore compare our estimates to the true model. The derivative was computed numerically (first order through finite differences) and the training samples were generated as follows: We chose $N_1 = 500$ samples from the reference function in the interval [-20, 20], $N_2 = 400$ samples from the derivative signal in the interval [-20, 0], and $N_3 = 400$ samples from the delayed signal in the interval [-20, 0]. All samples were randomly uniformly chosen in the intervals mentioned and Gaussian noised was added to yield realistic observations. The experiment then consisted in the reconstruction the reference signal in the interval [-20, 20], and the imputation of the derivative and delayed signals over the interval [0, 20].

Fig. 2 shows the ground truth and MOSM estimates for all three synthetic signals and the co-variances (normalised), and Table 1 reports the MAE for all models over ten realisations of the experiment. Notice that the proposed model successfully learnt all cross-covariances $\text{cov}(f(\cdot), f'(x))$ and $\text{cov}(f(x), f(x - \delta))$, and autocovariances without prior information about the delayed or the derivative relationship between the two channels. Furthermore, MOSM was the only model that successfully extrapolated the derivate signal and the delayed signal simultaneously, this is due the fact that the cross-covariances needed for this setting are not linear combinations of univariate kernels, hence models based on latent processes fail in this synthetic example.

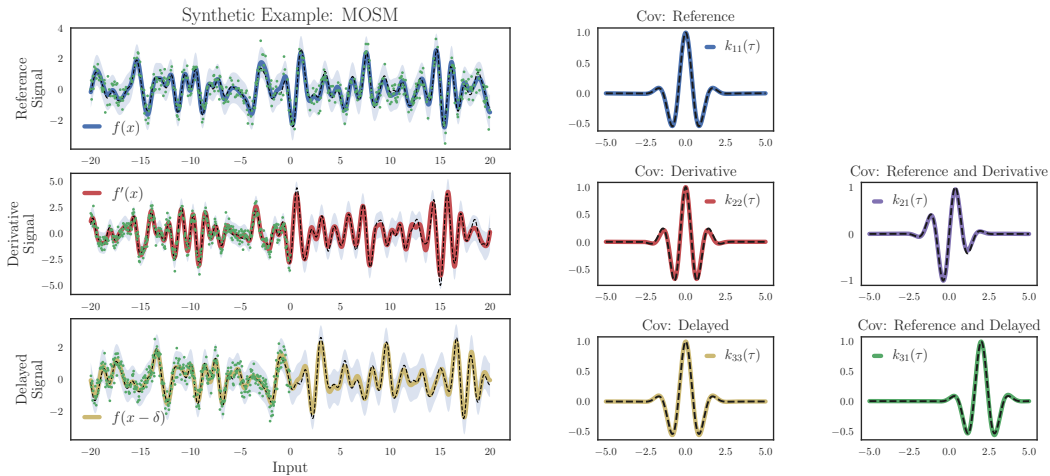

Figure 2: MOSM learning of the covariance functions of a synthetic reference signal, its derivative and a delayed version. **Left:** synthetic signals, **middle:** autocovariances, **right:** cross-covariances. The dashed line is the ground truth, the solid colour lines are the MOSM estimates, and the shaded area is the 95% confidence interval. The training points are shown in green.

Table 1: Reconstruction of a synthetic signal, its derivative and delayed version: Mean absolute error for all four models with one-standard-deviation error bars over ten realisations.

| Model | Reference | Derivative | Delayed |
|---|---|---|---|
| CONV | $0.211 \pm 0.085$ | $0.759 \pm 0.075$ | $0.524 \pm 0.097$ |
| SM-LMC | $0.166 \pm 0.009$ | $0.747 \pm 0.101$ | $0.398 \pm 0.042$ |
| CSM | $0.148 \pm 0.010$ | $0.262 \pm 0.032$ | $0.368 \pm 0.089$ |
| MOSM | $0.127 \pm 0.011$ | $0.223 \pm 0.015$ | $0.146 \pm 0.017$ |

## 4.2 Climate data

The first real-world dataset contained measurements[3] from a sensor network of four climate stations in the south on England: Cambermet, Chimet, Sotonmet and Bramblemet. We considered the normalised air temperature signal from 12 March, 2017 to 16 March, 2017, in 5-minute intervals (5692 samples), from where we randomly chose $N = 1000$ samples for training. Following [4], we simulated a sensor failure by removing the second half of the measurements for one sensor and leaving the remaining three sensors operating correctly; we reproduced the same setup across all four sensors thus producing four experiments. All models considered had five latent signals/spectral components.

For all four models considered, Fig. 3 shows the estimates of missing data for the Cambermet-failure case. Table 2 shows the mean absolute error for all models and failure cases over the missing data region. Observe how all models were able to capture the behaviour of the signal in the missing range, this is because the considered climate signals are very similar to one another. This shows that the MOSM can also collapse to models that share parameters across pairs of outputs when required.

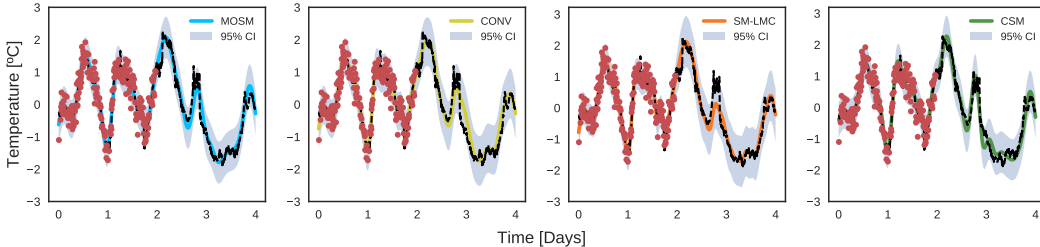

Figure 3: Imputation of the Cambermet sensor measurements using the remaining sensors. The red points denote the observations, the dashed black line the true signal, and the solid colour lines the predictive means. From left to right: MOSM, CONV, SM-LMC and CSM.

Table 2: Imputation of the climate sensor measurements using the remaining sensors. Mean absolute error for all four experiments with one-standard-deviation error bars over ten realisations.

| Model | Cambermet | Chimet | Sotonmet | Bramblemet |
|---|---|---|---|---|
| CONV | $0.098 \pm 0.008$ | $0.192 \pm 0.015$ | $0.211 \pm 0.038$ | $0.163 \pm 0.009$ |
| SM-LMC | $0.084 \pm 0.004$ | $0.176 \pm 0.003$ | $0.273 \pm 0.001$ | $0.134 \pm 0.002$ |
| CSM | $0.094 \pm 0.003$ | $0.129 \pm 0.004$ | $0.195 \pm 0.011$ | $0.130 \pm 0.004$ |
| MOSM | $0.097 \pm 0.006$ | $0.137 \pm 0.007$ | $0.162 \pm 0.011$ | $0.129 \pm 0.003$ |

These results do not show a significant difference between the proposed model and the latent processes based models. In order to test for statistical significance, the Kolmogorov-Smirnov test [20, Ch. 7] was used with a significance level $\alpha = 0.05$, concluding that for the Sotonmet sensor we can assure that the MOSM model yields the best results. Conversely, for the Cambermet, Chimet and Bramblemet sensors, MOSM and CSM provided similar results, though we cannot confirm their difference is statistically significant. However, given the high correlation of these signals and the

similarity between the MOSM model and the CSM model, the close performance of these two models on this dataset is to be expected.

### 4.3 Heavy metal concentration

The Jura dataset [3] contains, in addition to other geological data, the concentration of seven heavy metals in a region of $14.5$ km$^2$ of the Swiss Jura, and it is divided into a training set (259 locations) and a validation set (100 locations). We followed [3, 4], where the motivation was to aid the prediction of a variable that is expensive to measure by using abundant measurements of correlated variables which are less expensive to acquire. Specifically, we estimated Cadmium and Copper at the validation locations using measurements of related variables at the training and test locations: Nickel and Zinc for Cadmium; and Lead, Nickel and Zinc for Copper. The MAE—see eq. (13)—is shown in Table 3, where the results for the CONV model were obtained from [4] and all models considered five latent signals/spectral components, except for the independent Gaussian process (denoted IGP).

Observe how the proposed MOSM model outperforms all other models over the Cadmium data, which is statistical significant with a significance level $\alpha = 0.05$. Conversely, we cannot guarantee a statistically-significant difference between the CSM model and the MOSM in the Copper case. In both cases, testing for statistical significance against the CONV model was not possible since those results were obtained from [4]. On the other hand, the higher variability and non-Gaussianity of the Copper data may be the reason of why the simplest MOGP model (SM-LMC) achieves the best results.

Table 3: Mean absolute error for the estimation of Cadmium and Copper concentrations with one-standard-deviation error bars over ten repetitions of the experiment.

| Model | Cadmium | Copper |
|---|---|---|
| IGP | $0.56 \pm 0.005$ | $16.5 \pm 0.1$ |
| CONV | $0.443 \pm 0.006$ | $7.45 \pm 0.2$ |
| SM-LMC | $0.46 \pm 0.01$ | $7.0 \pm 0.1$ |
| CSM | $0.47 \pm 0.02$ | $7.4 \pm 0.3$ |
| MOSM | $0.43 \pm 0.01$ | $7.3 \pm 0.1$ |

## 5 Discussion

We have proposed the multioutput spectral mixture (MOSM) kernel to model rich relationships across multiple outputs within Gaussian processes regression models. This has been achieved by constructing a positive-definite matrix of complex-valued spectral densities, and then transforming them via the inverse Fourier transform according to Cramér's Theorem. The resulting kernel provides a clear interpretation from a spectral viewpoint, where each of its parameters can be identified with frequency, magnitude, phase and delay for a pair of channels. Furthermore, a key feature that is unique to the proposed kernel is the ability joint model delays and phase differences, this is possible due to the complex-valued model for the cross-spectral density considered and validated experimentally using a synthetic example—see Fig. 2. The MOSM kernel has also been compared against existing MOGP models on two real-world datasets, where the proposed model performed competitively in terms of the mean absolute error. Further research should point towards a sparse implementation of the proposed MOGP which can build on [4, 21] to design inducing variables that exploit the spectral content of the processes as in [22, 23].

### Acknowledgements

We thank Cristóbal Silva (Universidad de Chile) for useful recommendations about GPU implementation, Rasmus Bonnevie from the GPflow team for his assistance on the experimental MOGP module within GPflow, and the anonymous reviewers. This work was financially supported by Conicyt Basal-CMM.

## Footnotes

[1]A function $g(x)$ is said to be integrable if $\int_{\mathbb{R}^n} |g(x)| \mathrm{d}x < +\infty$

[2]The extension to arbitrary $Q$ will be presented at the end of this section.

[3]The data can be obtained from `www.cambermet.co.uk.` and the sites therein.

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
