[Reviews · NeurIPS 2017]

Reviewer 1



This paper presents a multi-output Gaussian process co-variance function that is defined in the spectral domain. The approach of designing co-variances in the spectral domain is an important contribution to GPs that I personally feel have not gotten the attention that it deserves. This paper extends this to the multi-output scenario. The paper is an extension of the work presented in [1]. However, considering the multi-output case the challenge is how to describe cross-covariance which is does not have to be positive-definite. In this paper the authors describes just such an approach in a manner that leads to the spectral mixture covariance for multi-output GPs. The paper is well written with a short and clear introduction to the problem and to GP fundamentals. The explanation of the spectral mixture kernel is clear and the extensions proposed are to my knowledge novel. The relationship to previous work is well explained and I believe that there is sufficient novelty in the work to justify publication. The experiments are sufficient for a paper such as this, they do not provide a lot of intuition however, especially the Jura data-set is hard to actually reason about. But, for a paper such as this I think the experimental evaluation is sufficient. -[1] Wilson, A. G., & Adams, R. P. (2013). Gaussian process kernels for pattern discovery and extrapolation. In , Proceedings of the 30th International Conference on Machine Learning, {ICML} 2013, Atlanta, GA, USA, 16-21 June 2013 (pp. 1067–1075). : JMLR.org.

Reviewer 2



The paper formulates a new type of covariance function for multi-output Gaussian processes. Instead of directly specifying the cross-covariance functions, the authors propose to specify the cross-power spectral densities and using the inverse Fourier transform to find the corresponding cross-covariances. The paper follows a similar idea to the one proposed by [6] for single output Gaussian processes, where the power spectral density was represented through a mixture of Gaussians and then back-transformed to obtaining a powerful kernel function with extrapolation abilities. The authors applied the newly established covariance for a synthetic data example, and for two real data examples, the weather dataset and the Swiss Jura dataset for metal concentrations. Extending the idea from [6] sounds like a clever way to build new types of covariance functions for multiple-outputs. One of the motivations in [6] was to build a kernel function with the ability to perform extrapolation. It would have been interesting to provide this covariance function for multiple-outputs with the same ability. The experimental section only provides results with very common datasets, and the results obtained are similar to the ones obtained with other covariance functions for multiple-outputs. The impact of the paper could be increased by including experiments in datasets for which none of the other models are suitable. Did you test for the statistical significance of the results in Tables 1 and 2. It seems that, within the standard deviation, almost all models provide a similar performance.

Reviewer 3



The paper introduces a new covariance structure for multioutputs GPs, which corresponds to the generalisation of the spectral approach (Bochner Theorem) to build kernels from A. G. Adams and R. P. Adams. One particular asset of the proposed method is that the parameters of the model can be interpreted (such as delay between outputs or phase difference). The definition of covariance structures for multioutputs GPs is a challenging topic and relatively few methods are currently available to do so. The method proposed by the authors is theoretically sound and its effectiveness is demonstrated on several datasets. The method is clearly described in a well written paper, and the choices for the illustrations appear relevant to me. If the parameter learning isn't too troublesome (the paper does not contain much informations regarding this), I would expect this method to become a standard for multioutput GPs. Questions * How does the method behaves when few training points are available? * (more a curiosity) A channel with a large length-scale cannot be highly correlated with a channel with a small length-scales. How does that appear in your model? Comments * In the synthetic example (Section 4.1), the number of training points should be specified Minor remarks: I guess a word is missing in lines 74 and 140.